# Analysis of the Activity and Expression of Cyclooxygenases COX1 and COX2 in THP-1 Monocytes and Macrophages Cultured with Biodentine^TM^ Silicate Cement

**DOI:** 10.3390/ijms21062237

**Published:** 2020-03-24

**Authors:** Katarzyna Barczak, Mirona Palczewska-Komsa, Alicja Nowicka, Dariusz Chlubek, Jadwiga Buczkowska-Radlińska

**Affiliations:** 1Department of Conservative Dentistry and Endodontics, Pomeranian Medical University, Powstańców Wlkp 72, 70-111 Szczecin, Poland; mpalczewskakomsa@op.pl (M.P.-K.); nowicka6@gmail.com (A.N.); zstzach@pum.edu.pl (J.B.-R.); 2Department of Biochemistry and Medical Chemistry, Pomeranian Medical University, Powstańców Wlkp, 70-111 Szczecin, Poland; dchlubek@pum.edu.pl

**Keywords:** Biodentine^TM^, bioactive calcium-silicate cement, THP-1 monocytes, macrophages, cyclooxygenase-1 (COX-1), cyclooxygenase-2 (COX-2), inflammatory reaction

## Abstract

Biodentine^TM^ is a material based on hydrated calcium silicate with odontotropic properties. However, from the clinician’s perspective, every material used to fill a tooth—even those showing the optimal biochemical parameters—is in fact a foreign body introduced to the organism of the host. Therefore, apart from the chemical parameters of such materials, equally important is the so-called biocompatibility of such materials. The aim of the study was to investigate whether Biodentine^TM^, used in the regeneration of the pulp-dentine complex, may affect the expression of the enzymes cyclooxygenase 1 (COX1) and cyclooxygenase 2 (COX2) in THP-1 monocytes/macrophages and the amount of prostanoids synthesized by these enzymes-precursors of biologically active prostanoids such as prostaglandin E2 (PGE2) and thromboxane (TXB2) which are mediators of inflammation. An original aspect of this research is the use of the THP-1 monocyte/macrophage cell model and the use of biomaterial in direct contact with cells. In this way we tried to reflect the clinical conditions of regenerative pulp and periodontal tissue treatment using Biodentine^TM^. The results of our study showed a lack of macrophage activation (measured by flow cytometry) and a lack of stimulation of the expression of the studied cyclooxygenase enzymes (measured by Western blotting and fluorescent microscopy), as well as a lack of increase in the concentration (measured by ELISA method) of their inflammatory mediators (PGE2 and TXB2) in vitro incubated with Biodentine^TM^.

## 1. Introduction

Calcium Silicate-based Cements, for example Biodentine^TM^, BioAggregate, Theracal LC, Calcium Enriched Matrix (CEM), Endo Sequence Root Repair Material (ERRM) and iRoot BP Plus, MTA (Mineral Trioxide Aggregate), NeoMTA plus, MTA Repear HP—used in modern dentistry—are classified as bioinert, bioactive and biodegradable materials. The shared feature of these bioceramic materials is that they are especially developed to perform the intended function, that is, they act as a root canal sealant, cements and materials used for root treatment and filling. Moreover, these materials can be used in cases of pulp exposure due to injury, caries or other mechanical causes in the form of direct pulp capping [1,2,3]. It is proved that calcium-silicate cements used for direct pulp capping affect the initial inflammatory stages and induce the healing process [4]. The currently used MTA is considered the golden standard in the biological treatment of the pulp [5].

Biodentine^TM^ (Septodont, Saint Maur des Fsses, France) is an example of a material based on hydrated calcium silicate. It shows odontotropic properties, that is, it stimulates the formation of reactive and reparative dentin and helps maintain proper sensitivity of the dental pulp [6,7,8,9]. The efficacy of the biological treatment of the pulp is higher in the younger patients as compared to the older patients [8]. This material is also used for sealing root perforation [9,10]. The teeth with perforations treated with Biodentine^TM^ show regression of the inflammatory process and reduction in the bone resorption over time [1]. Moreover, it may also stimulate the expression of cell differentiation factors (including the osterix) and promote osteoblast differentiation and, consequently, bone neoformation [9,11]. Furthermore, it promotes fibroblast and osteoblast differentiation, stimulating the formation of collagen bundles of periodontal ligament and bone matrix of the alveolar process, respectively, favoring periodontal tissue repair [9,11]. In vivo studies have reported that Biodentine^TM^ exhibits low cytotoxicity in cultures of osteoblasts [12,13]. In pulp tissue, it induces cell proliferation and the expression of dentine sialoprotein and osteopontin [14]. There are reports on the use of Biodentine^TM^ in pulpotomy procedures [15]. It was shown that Biodentine^TM^ stimulates the formation and mineralization of the tissue barrier in the dental pulp [16]. However, from the clinician’s perspective, every material used to fill a tooth, even those showing the optimal chemical parameters, is in fact a foreign body introduced to the organism of the host. Therefore, apart from the chemical parameters of such materials, equally important is the so-called biocompatibility of such materials [1,5,7,8].

The factors underlying pulp disease may, as a result, lead to premature tooth loss. A tooth with a healthy pulp is more likely to be kept longer in the oral cavity. Due to its structure, the pulp has limited reparative abilities. Two dental tissues that is, dentine and pulp, comprise a structural-functional unit also known as the pulp and dentin complex. The two tissues affect one another as the pulp provides nutrients to the dentine and the dentine serves as a protective barrier (Figure 1). As a result of this, the pulp’s functional status as well as the integrity of the barrier preventing the entry of bacteria are preserved. Additionally, the pulp-dentin complex is a basic unit for the assessment of the biological risk connected with applying materials to the exposed pulp during the biological treatment [7,8]. 

The pulp of a healthy tooth has cells involved in immune reactions such as macrophages, lymphocytes, plasma cells and mast cells. The macrophages origin from the bloodstream monocytes, are spread throughout the pulp but usually they are located in a small distance from the walls of the blood vessels and fine capillaries. Under physiological conditions, macrophages are present in the pulp in a dormant state. Only during the development of the inflammatory process, do they show mobility and move to the site of inflammation [3,4,5,9].

Nowadays, the treatment of deep caries involves the use of materials with increasingly wider and better biochemical properties—desirable qualities in the production of dentin substitutes. However, despite a continuous introduction of new materials and techniques, some side effects can still be observed. One such side effect is an immune response that can lead to increased inflammation. Many studies show that use of biomaterials may cause the release of cytokines and chemokines, along with activation of the complement system, thereby increasing the intensity of inflammation which may significantly shorten the life of the tooth in the oral cavity and, in extreme cases, lead to the necessity of its premature removal [4,5,7,8].

Monocytes/macrophages are one of the first cells to come into contact with the filling material, just after it is placed on the exposed or partially damaged pulp [17]. They show the ability to synthesize and produce cytokines, therefore constituting a significant source [18]. This makes them an important factor in the initiation of both destructive as well as repair processes [19]. The inflammation is strongly related to the activity of macrophages, the cells taking part in the organism’s reaction to various factors, including the filling materials [20,21]. Due to secreting cytokines and growth factors, they initiate the reparative as well as destructive processes [19,22,23]. The initial phase after implantation includes intensive secretion of pro-inflammatory cytokines such as IL-1β, IL-6, TNF-α and chemokines MCP-1 and MIP-1α (monocyte chemoattractant protein-1 and macrophage inflammatory protein-1α [19,23,24]. In vitro macrophage cultures also showed an increase in the expression and activity of cyclooxygenase1 (COX1) and cyclooxygenase 2 (COX2), enzymes catalyzing the conversion of arachidonic acid into prostaglandin H2 (PGH2) [25,26]. This conversion consists of two stages, with the resulting PGH2 the precursor of biologically active prostanoids such as prostaglandin E2 (PGE2) and thromboxane A2 (TXA2) [27]. Until recently, COX-1 has been considered a constitutive enzyme which does not play a significant role in the inflammatory process [28]. However, it is now known that in some tissues this enzyme is inducible and plays a role in the initial phase of the response to the factors initiating prostanoid synthesis. COX-1 is a source of PGE2 and TXA2 (which is metabolized to the more stable TXB2). COX-2 is an enzyme subject to inducible expression in response to for example, pro-inflammatory cytokines or cytokines produced by cell growth factors (including monocytes) and plays a dominant role in chronic inflammation [29,30].

The aim of this study was to investigate whether Biodentine^TM^—a silicate based material used in the regeneration of the pulp-dentine complex, may affect the expression of the enzymes COX1 and COX2 in THP-1 monocytes/macrophages, as well as the amount of prostanoids synthesized by these enzymes (PGE2 and TXB2)—the mediators of inflammation.

An original aspect of this research is the use of the THP-1 monocyte/macrophage cell model and the use of biomaterial in direct contact with cells. In this way we tried to reflect the clinical conditions of regenerative pulp and periodontal tissue treatment using Biodentine^TM^.

## 2. Results

### 2.1. Biodentine^TM^ -Induced Activation of THP-1 Monocytes

Biodentine^TM^-induced activation of THP-1 monocytes showed that CD68 expression (marker for macrophages differentiation) increased after 48 h treatment with tested material as compared to non-treated cells. Expression of CD14 (marker for monocyte differentiation) 0.6% did not change (Figure 2).

### 2.2. THP-1 Monocytes and Macrophages Culture Visualization

Cultures of monocytes and macrophages under control conditions for 24 h and 48 h (RPMI medium with 10% FBS), as well as incubated with Biodentine^TM^ for 24 h and 48 h, showed normal morphology (Figure 3 and Figure 4). 

### 2.3. Prostaglandin E2 (PGE2) in THP-1 Monocytes and Macrophages

Incubation of THP-1 monocytes for 24 h under Biodentine24 conditions caused a statistically significant increase in the concentration of PGE2 in the medium in relation to the cells incubated under control (control 24), (by about 122%, *p* = 0.002). Extension of the incubation time from 24 h to 48 h caused a significant increase in the concentration of PGE2 in the culture with Biodentine48 as compared with the control 48 (by about 70%, *p* = 0.026), (Figure 5A). 

The cultures of macrophages incubated with Biodentine for 24 h showed a statistically significant increase in PGE2 concentration (more than 7 times, *p* = 0.0022) in comparison to incubated cells under control conditions (control 24). Extension of incubation time increased the concentration of PGE2 released by cells under control conditions (control 24) in comparison to control 48 (over 7 times, *p* = 0.0022), (Figure 5B).

### 2.4. Thromboxane TXB2 in THP-1 Monocytes and Macrophages

In THP-1 monocytes incubated for 24 h with Biodentine^TM^, a statistically significant increase in TXB2 concentration was found (by about 100%, *p* = 0.009) as compared with control 24. Extension of the incubation time of cells cultured under control conditions resulted in a significant increase of TXB2 concentration in the medium (by about 170%, *p* = 0.004) as compared to control 24, (Figure 6A). A statistically significant increase in TXB2 was observed in control 48 as compared to control 24 (5-fold, *p* = 0.05), (Figure 6B).

### 2.5. Cyclooxygenase-1 Expression in Monocytes

In THP-1 monocytes incubated for 24 h with Biodentine^TM^ no significantly higher COX-1 expression was found as compared to control 24. Prolongation of the incubation time of cells cultured under control conditions to 48 h resulted in an increase in enzyme expression but also not statistically significant as compared to control 24. Prolongation of the incubation time of monocytes cultured with Biodentine^TM^ to 48 h increased COX-1 expression (by approx. 67%) when compared to control 48, however, it was not statistically significant (Figure 7A,B). No visible differences in enzyme protein expression were observed in the image under the confocal microscope (Figure 8).

### 2.6. Cyclooxygenase-2 Expression in Macrophages

Macrophages incubated for 24 h with Biodentine^TM^ did not exhibit significantly higher COX-2 expression as compared to control 24. Prolongation of the incubation time of the control cells to 48 h resulted in an increase in enzyme expression, yet also not statistically significant in comparison to control 24. Prolongation of the incubation time of macrophages incubated with Biodentine^TM^ to 48 h resulted in an increase in COX-2 expression (by approx. 36%) in comparison to control 48 but the observed increase in expression was not statistically significant, (Figure 9A,B). No visible differences in enzyme protein expression were observed in the image from the confocal microscope (Figure 10).

## 3. Discussion

The advantage of biomaterials containing mineral trioxide and calcium silicate consists in their potential bioactive properties [1,2,3,5,7,8,31]. Bioactivity can be defined as materials that elicit a specific biological response at the interface between tissues and the material, which results in the formation of a bond [32]. Similarly to other silicate-containing materials, Ca(OH)_2_ is produced upon reaction with water, leading to a high alkaline pH that activates and stimulates the expression of alkaline phosphatase, favoring the formation of mineralized tissue and having an antimicrobial effect. In addition, alkaline pH can neutralize lactic acid from osteoclasts and prevent dissolution of the mineralized components of teeth [33]. For many decades, calcium hydroxide was the material of choice among the various available pulp-capping agents [8]. However, there are shortcomings when using this material such as its dissolution in tissue fluids and degradation on tooth flexure, the formation of tunnel defects beneath dentinal bridges and poor sealing. The use of calcium silicate-based cements (biomaterials with calcium oxide and carbonate filler additives) in dentistry became a treatment of choice used for the purpose of establishing dental bridge in the procedure of direct pulp capping [5,7,8,34,35]. Studies have shown that mineral trioxide aggregate (MTA) and Biodentine^TM^ may be used as an alternative to Ca(OH)_2_ for treating pulp wounds. MTA stimulates formation of dentin bridges faster than calcium hydroxide. However, MTA is reportedly difficult to use because of its long setting time, poor handling properties, high material costs and the discoloration potential of dental tissue. Biodentine^TM^ presents adequate biological response in vivo, similar to MTA [9]. Biomaterials should fill and rebuild the tooth, but they should also be biocompatible, bioactive and should not cause inflammatory reaction of the tooth pulp. So far, there have been few studies in which this material has been studied in in vitro cell cultures [36], however there are no reports of the effect of Biodentine^TM^ on the metabolism of monocytes/macrophages in direct contact with this material in the context of the participation of these cells in the initiation and propagation of inflammatory reaction of the pulp. In our study, 48 h of incubation of THP-1 monocytic cells with Biodentine^TM^ did not cause a significant increase in the concentration of TXB2 or PGE2 in the medium as compared to the cells incubated in control conditions. The presence of Biodentine^TM^ also did not affect the activation of THP-1 monocytes and their transformation into macrophages after this time. The obtained results suggest that this material does not increase COX-2 activity and therefore does not stimulate inflammatory reactions. Interestingly, in the case of PGE2 concentration in macrophages incubated with Biodentine^TM^ for 48 h, we found a statistically insignificant lower concentration of this prostaglandin in relation to control. These findings confirm the suggestions of the anti-inflammatory properties of silicate base material [36]. Our results also indirectly confirm the observations of Loison-Robert et al. (2018), [36] whose in vitro studies conducted on human cells and dental stamina cells, incubated as in our model directly from Biodentine^TM,^ showed a lack of inflammation induction. They also showed that the incubation of Biodentine cells showed no significant cytotoxicity but some cells showed apoptosis in direct contact. Increased induction of mineralization in the presence of Biodentine^TM^ was also observed. This differentiation was accompanied by expression of odontoblast-associated genes. Also, other in vitro studies, enumerate the positive aspects of Biodentine^TM^, such as low cytotoxicity [37], good biocompatibility, bioactivity [1,2,3,5,7,8,31,38] and biomineralization [7,8,39]. Some in vitro studies have shown better sealing ability [40], higher mechanical compressive strength and microhardness [38,41] when compared to other bioceramic materials, suggesting that it may be used as an effective temporary sealant [42]. Additionally, Da Fonesca et al. (2016) and De Sousa Reis et al. (2019) stated that the significant regression of inflammatory reaction allows a conclusion that Biodentine^TM^ is a biocompatible material [11,43]. In an odontoblast-like mouse cell line (MDPC-23), Paula et al. (2019) [44] conducted an assessment of the cytotoxicity and bioactivity of three different direct pulp capping materials, calcium hydroxide (Life^®^), mineral trioxide aggregate (WhiteProRoot^®^MTA) and calcium silicate (Biodentine™). The authors observed that metabolic activity and cellular viability decreased due to calcium hydroxide-based cements. There was also a marked increase in cell death and some notable changes of the cell cycle. No protein synthesis or formation of calcium nodules was observed. Owing to mineral trioxide aggregate sand tricalcium silicates materials, metabolic activity and cell viability increased. Additionally, the percentage of living cells was high and there were no interferences with the cell cycle. In the subsequent stages of differentiation as well as mineralization, a better performance was found for tricalcium silicate cements. Alkaline phosphatase expression increased significantly, as did dentin sialoprotein and formation of calcium nodule—as compared to the mineral trioxide aggregate cements. The study referred to supports the stipulation that it is advisable to use tricalcium silicate as well as mineral aggregate trioxide cement-based for pulp capping procedures [44]. The results of our study showed a lack of macrophage activation and a lack of stimulation of the activity of the tested cyclooxygenase enzymes, as well as a lack of increase in the concentration of their inflammatory mediators (PGE2 and TXB2) under in vitro incubation with Biodentine^TM^. The results of our study showed that macrophages incubated with Biodentine in vitro in 48 h did not show activation characteristics. Moreover, no significant increase in the expression of cyclooxygenase or significant increase in inflammation mediators (PGE2 and TXB2) were found. This indicates that Biodentine, under the conditions presented in the paper, does not initiate inflammation and does not contribute its propagation. The same model of in vitro studies was used by Sikora et al. to determine how the production of PGE₂ and TXB₂ in THP-1 monocytes/macrophages was influenced by the titanium 3D plates and dedicated screws [45,46]. With the exception of the present study, there are no studies in the literature concerning the direct effect of exposure to Biodentine^TM^ on the expression and activity of cyclooxygenases and the production of PGE2 and TXB2 in monocytes and macrophages of the THP-1 cell line. Monocytes/macrophages are one of the first cells to come into contact with the filling material after its introduction [18]. At the same time, having the ability to synthesize and release pro-inflammatory factors, they are their important source, hence their role in the initiation of reparative and destructive processes.

The aim of the manufacturers of the materials used in regenerative endodontics is to create a product that does not cause excessive reaction in the surrounding tissues. Such a reaction would prevent the integration of the bulking material into the tissues of the host. Therefore, the ability of a good bulking material to stimulate macrophages, which are the key cells in early immune response and the development of inflammation [47], is considered to be one of the most important features determining biocompatibility. This data is in line with publications depicting Biodentine^TM^ as a biocompatible material, causing a negligible local immune response [5,7,8,9,11,48]. The results of the present study suggest that Biodentine^TM^ seems to be a biocompatible material which does not stimulate the pro-inflammatory reaction.

## 4. Materials and Methods

### 4.1. Reagents

THP-1 monocytic cells were obtained from American Type Culture Collection (ATCC, Rockville, MA, USA). The THP-1 cellular line monocytes and macrophages obtained from them constitute a widely applied cellular model used to investigate immune reaction. It was proven that the results of studies conducted with these cells can be applicable with respect to the human organism. THP-1 show similarities to the human monocytes/macrophages both in terms of morphology as well as function [48]. The literature on the subject considers them to be an appropriate model for the study of monocyte/macrophage response, for the purpose of determining “macrophage differentiation,’ as well as investigating the effect of external factors on macrophages [49]. RPMI medium, glutamine and antibiotics (penicillin and streptomycin), phosphate buffered saline (PBS), phorbol 12-myristate 13-acetate (PMA) were purchased from Sigma–Aldrich (Poznań, Poland). Fetal bovine serum was purchased from Gibco (Gibco, Paisley, UK). Bakerbond columns were obtained from Witko Group, Poland. Prostaglandin E2 EIA Kit and Thromboxane B2 EIA Kit were purchased from Cayman, USA; Micro BCA Protein Assay kit was purchased from Thermo Scientific (Poznań, Poland). The primary monoclonal antibodies against COX-1, COX-2 and ß-actin were purchased from Santa Cruz Biotechnology, Heidelberg, Germany). The secondary antibodies (goat anti-mouse IgG HRP) were obtained from Santa Cruz Biotechnology (Heidelberg, Germany). Biodentine^TM^ was obtained from Septodont, Saint Maur des Fsses, France.

### 4.2. Chemical Properties and Preparation of Biodentine^TM^

Biodentine^TM^ (Septodont, Saint Maur des Fosses, France) occurs in the form of capsules with powder and ampules with liquid. The powder consists mainly of tricalcium and dicalcium silicates (3CaO**·**SiO_2_) and 2CaO_8_SiO_2_), which are the main constituents of Portland cement and calcium carbonate (CaCO_3_). Zirconium dioxide (ZrO_2_) is an additive that ensures good visibility in the x-ray images. The liquid is an aqueous solution of calcium chloride (CaCl_2_**·**2H_2_O) with an admixture of polycarboxylate [7,8,10]. In the combination of powder and liquid, a hydrated calcium silicate (so-called CSH gel–hydrated calcium silate gel) and calcium hydroxide are obtained. The catalyst of the reaction is calcium chloride and the tricalcium silicate is responsible for the binding reaction. Calcium carbonate acts as a filler which improves the mechanical properties of the material [7,8,10]. The binding time of Biodentine^TM^ is about 12 min from the time of its mixing. After mixing the powder and the liquid, the whole cavity in the tooth can be filled with the preparation obtained.

Biodentine^TM^ was prepared according to the manufacturer’s instructions under aseptic conditions and transformed into powder by grinding under cold conditions (−20 °C) and then sterilized by dry heat. The ground powder was added to the medium to prepare stock solutions (10 mg/mL) of material and vortexed, a final concentration of 2 mg/mL was used. Table 1 present detailed manufacturer data on the Biodentine^TM^ used for testing.

### 4.3. Cell Culture and Treatment

The experiments were conducted on human macrophages obtained from THP-1 cells. The cells were cultured in Roswell Park Memorial Institute (RPMI) 1640 (Sigma, St. Louis, MO, USA), supplemented with 100 IU/mL penicillin and 10 µg/mL streptomycin (Life Technologies, Inc., GrandIsland, NY, USA) in the presence of 10% thermally inactivated fetal bovine serum (FBS, LifeTechnologies, Poznań, Poland). Cells were cultured in a humid atmosphere at 37 °C in 5% CO_2_, the medium was refreshed every 48 h. Prior to the experiment, THP-1 cells were placed in culture flasks at an initial density of 2.5 × 10^5^ cells/well (Corning, Cambridge, MA, USA). The differentiation of THP-1 cells in the macrophages was achieved by administration of 100 nM PMA for 24 h [50].

### 4.4. Verification of Biodentine^TM^ Induced Activation of THP-1 Monocytes and Initiation of Inflammatory Reaction

In the first experiment, THP-1 cells were cultured for 24 and 48 h in RPMI medium with 10% FBS in the presence of Biodentine^TM^ prepared according to the manufacturer’s instructions. After incubation, the cells were collected by scraping and the pellets were obtained by centrifugation (800 G, 10 min). Then, cold PBS was added to the cell pellets and the samples were frozen at −80 °C for further analysis. Protein concentration was measured with a Micro BCA Protein Assay Kit (Thermo Scientific, Rockford, IL, USA). The remaining supernatants were placed in new tubes and stored at −80 °C until further analysis, that is, extraction and measurement of PGE2 and TXB2 concentration by ELISA method and COX-1 and COX-2 expression changes by Western blotting method. Differentiation of THP-1 monocytic cells in THP-1 macrophages (activation of THP-1 monocytic cells) was measured by flow cytometry. The in vitro culture model applied in the course of the present study has already been used previously for the purpose of the clinical verification of inducing inflammatory reaction of 3D plates used for surgical treatment of condylar fractures of facial bones [45].

### 4.5. The Differentiation of THP-1 Cells into Macrophages. Flow Cytometry Measurement

The activity of THP-1 monocytic cells (their differentiation in macrophages without PMA but only in the presence of Biodentine^TM^ as in the first experiment) was checked by the expression of CD 14 and CD 68 antibodies, as evaluated by flow cytometry using mouse antibody against human CD14 FITC and mouse antibody against human CD64 Alex Fluor 647 clone Y1: 82A (BD Pharmingen San Diego, CA, USA). Cells were compared to isotype control, IgG1 κ mouse and IgG2b κ mouse (BD Pharmingen). Procedure in brief—the cells were stained in phosphate buffered saline (PBS, Ca^2+^ and Mg^2+^) supplemented with 2% bovine calf serum (BCS, Hyclone, Logan, UT, USA). After the last wash, the cells were resuspended in PBS and were analyzed with FACS using a Navios flow cytometer (Beckman Coulter, Brea, CA, USA).

### 4.6. Verification of Biodentine^TM^ -Induced Initiation of the Inflammatory Reaction in Macrophages

In the second experiment, THP-1 macrophages were cultured for both 24 h and 48 h under the same conditions in the presence of Biodentine^TM^ as in the first experiment. After incubation, the cells were collected as before and the protein concentration was measured as in the first experiment using a Micro BCA Protein Assay Kit (Thermo Scientific); PGE2 and TXB2 were measured by ELISA, COX-1 and COX-2 expression by Western blotting.

### 4.7. Measurements of COX-1 and COX-2 Activity

The activity of COX-1 and COX-2 cyclooxygenase were tested in vitro by quantitative measurements of their products, PGE2 and thromboxane A2 (TXA2). Cells were incubated for 48 h with Biodentine^TM^, as described above. PGE2 and TXA2 were extracted from culture supernatants using Bakerbond SPE columns (J.T. Baker, USA) as described in the manufacturer’s manual. The concentration of PGE2 released was measured spectro-photometrically with an enzyme immunoassay kit (Cayman Chemical, Warsaw, Poland) according to the manufacturer’s protocol. Since TXA2 has a short half-life (37 s) and is rapidly hydrolyzed non-enzymatically to its stable thromboxane B2 derivative (TXB2), a thromboxane B2 enzyme immunoassay kit (Cayman Chemical, Warsaw, Poland) was used to measure free TXA2 indirectly.

### 4.8. Western Blotting Analysis of COX-1 and COX-2 Expression

Cells incubated with Biodentine^TM^ were removed from the PBS. After scraping, they were lysed with protease inhibitor lysis buffer, ethylene-diaminetetra-acetic acid 5 mM, sodium dichloroisocyanurate 1%, TRITON-X 1% and equal amounts of protein were separated in gel electrophoresis and transferred to a nitrocellulose membrane (Thermo Scientific, Pierce Biotechnology, Poznań, Poland) at 157 mA for 2 h at room temperature. After blocking the membrane of 5% non-fat milk in Tris-buffered saline (Sigma-Aldrich, Poznań, Poland) containing 0.1% Tween 20 (Sigma-Aldrich, Poland) for the whole night at 4 °C and incubated with primary monoclonal antibodies direct against COX-1 or COX-2 (Santa Cruz Biotechnology, Heidelberg, Germany) in dilution 1:200 with a monoclonal anti-ß-actin (1:5000; Santa Cruz Biotechnology, Heidelberg, Germany) and next with secondary antibodies (goat anti-mouse IgG HRP; Santa Cruz Biotechnology, Heidelberg, Germany) in dilution 1:5000. Signals were visualized by chemiluminescence (Thermo Scientific, Pierce Biotechnology, Poznań, Polad). ImageJ 1.41o (NIH, Madison, WI, USA) was used for densitometric analysis of bands.

### 4.9. Imaging of Cyclooxygenase-1 and Cyclooxygenase-2 Expression

Cells were incubated with Biodentine^TM^ on microscope slides according to the procedure mentioned above. After the end of cultivation, the cells were rinsed with PBS and fixed with 4% buffered formalin for 15 min at room temperature.

### 4.10. Protein Concentration

Protein concentration was measured by a MicroBCA Protein Assay Kit (Thermo Scientific, Pierce Biotechnology, USA) using a spectrophotometer (UVM340, ASYS). The Thermo Scientific™ Micro BCA™ Protein Assay Kit is a detergent-compatible bicinchoninic acid formulation for the colorimetric detection and quantitation of total protein. An adaptation of the Thermo Scientific™ BCA Protein Assay Kit (Product No. 23225), the Micro BCA Kit has been optimized for use with dilute protein samples (0.5–20 µg/mL). The unique patented method uses bicinchoninic acid (BCA) as the detection reagent for Cu+1, which is formed when Cu+2 is reduced by protein in an alkaline environment. A purple-colored reaction product is formed by the chelation of two molecules of BCA with one cuprous ion (Cu+1). This water-soluble complex exhibits a strong absorbance at 562 nm that is linear with increasing protein concentrations. The result is an extremely sensitive colorimetric protein assay in a test tube or microplate assay format.

### 4.11. Statistical Analysis

The statistical analysis of obtained results was conducted using Statistica 10 software (Statsoft, Poland). The results were expressed as arithmetical mean ± standard deviation (SD). The distribution of variables was evaluated using Shapiro-Wilk *W*-test. The nonparametric tests were used for further analyses, because distribution in most cases deviated from normal distribution. The results were subjected to U Mann test. The level of significance was set at *p* < 0.05.

## 5. Conclusions

Biodentine^TM^ used in vitro does not increase the expression of COX1 and COX2 enzymes in THP-1 monocytes/macrophages and the prostanoids PGE2 and TXB2 mediators of inflammation synthesized by these enzymes.

## Figures and Tables

**Figure 1 ijms-21-02237-f001:**
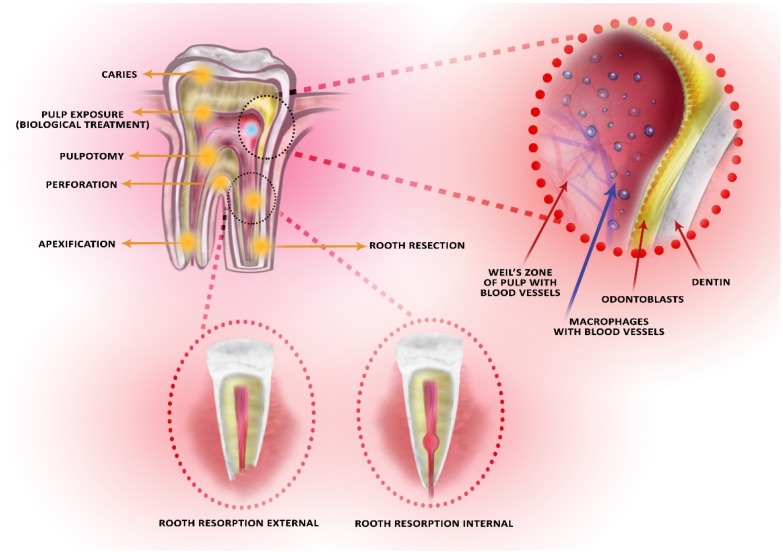
The site of clinical application of Biodentine^TM^ and the location of the cells taking part in the inflammatory reaction. The black dotted circles indicate the enlargement of the location of the external and internal rooth resorption.

**Figure 2 ijms-21-02237-f002:**
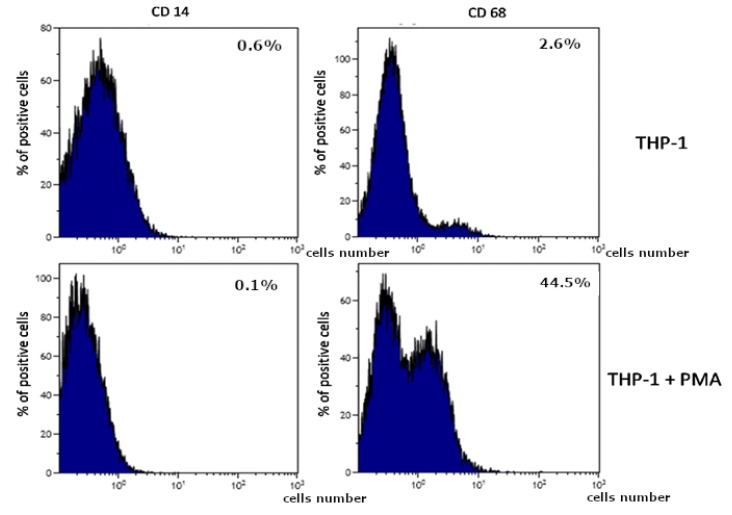
The effect of Biodentine^TM^ on the activation of THP-1 monocytes. THP-1 monocytic cells were incubated in the presence of Biodentine without phorbol 12-myristate 13-acetate (PMA (**upper quadrants**) and treated with PMA (**lower quadrants**). Expression was determined by flow cytometry. CD68 expression (marker for macrophage differentiation) increased after 48 h treatment with PMA (200 nM) as compared to non-treated cells. Expression of CD14 (marker for monocyte differentiation) did not change significantly, *p* = 0.65 (U Mann-Whitney test). cells number.

**Figure 3 ijms-21-02237-f003:**
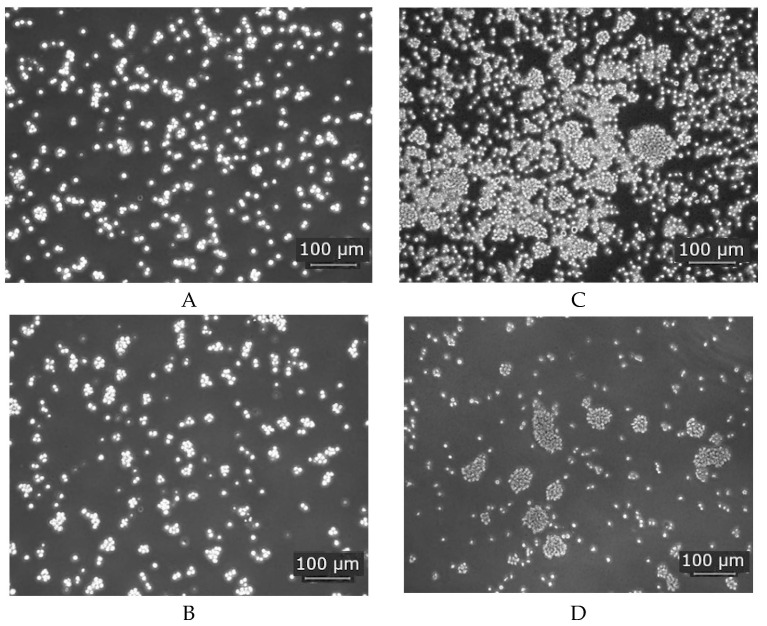
Imaging of THP1-monocytes cultures. Cells were cultured in RPMI medium with 10% FBS -control condition for 24 h (**A**) and control condition for 48 h (**B**); THP1-monocytes cultured in RPMI medium with 10% Fetal Bovine Serum (FBS) with Biodentine^TM^ for 24 h (**C**) and for 48 h (**D**). Normal cells are visible. Experiments were conducted as six separate assays (each assay in three replicates).

**Figure 4 ijms-21-02237-f004:**
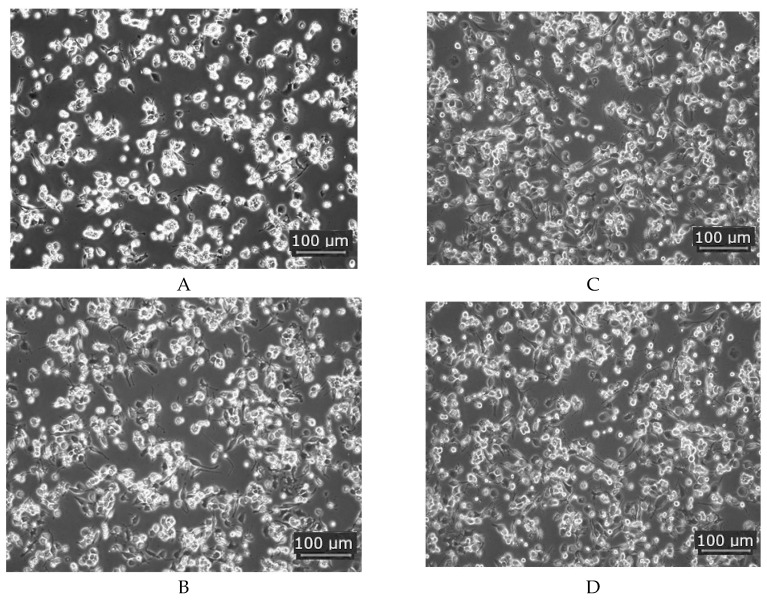
Imaging of macrophages culture. Cells were cultured in RPMI medium with 10% FBS-control condition for 24 h (**A**) and for control condition for 48 h (**B**); THP1-monocytes cultured in RPMI medium with 10% Fetal Bovine Serum (FBS) with Biodentine^TM^ for 24 h (**C**) and for 48 h (**D**). Normal cells are visible. Experiments were conducted as six separate assays (each assay in three replicates).

**Figure 5 ijms-21-02237-f005:**
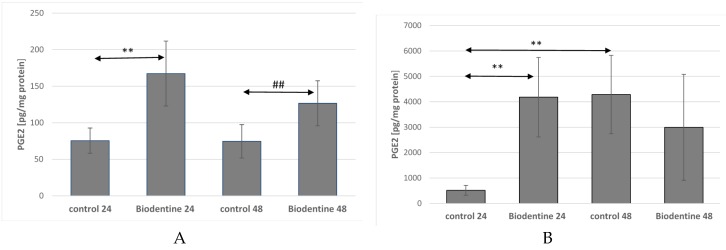
The concentration of prostaglandin E2 (PGE2) in THP-1 monocytes (**A**) and macrophages (**B**) cultured with Biodentine^TM^ Cells were cultured for 24 h and 48 h in RPMI medium with 10% FBS with Biodentine^TM^, after incubation the cells were harvested by scraping and PGE2 concentration was measured by ELISA method. Control cells were incubated in RPMI medium with 10% FBS. Experiments were conducted as six separate assays (each assay in three replicates). ** Statistically significant differences in comparison to control 24 (** *p* ≤ 0.05). ## Statistically significant differences in comparison to control 24 (## *p* ≤ 0.05), (U-Mann-Whitney test).

**Figure 6 ijms-21-02237-f006:**
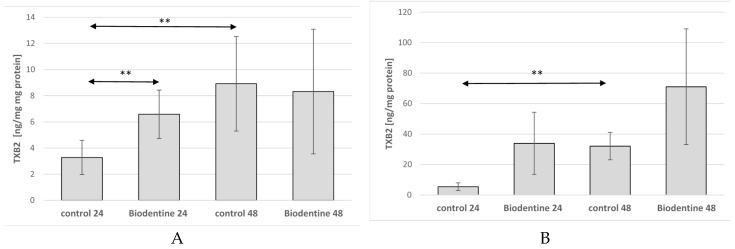
The concentration of thromboxane TXB2 (TXB2) in THP-1 monocytes (**A**) and macrophages (**B**) cultured with Biodentine^TM^. Cells were cultured for 24 h and 48 h in RPMI medium with 10% FBS with Biodentine^TM^, after incubation the cells were harvested by scraping and TXB2 concentration was measured by ELISA. Control cells were incubated in RPMI medium with 10% FBS. Experiments were conducted as six separate assays (each assay in three replicates). ** Statistically significant differences in comparison to control 24 (*p* ≤ 0.05), (U-Mann-Whitney test).

**Figure 7 ijms-21-02237-f007:**
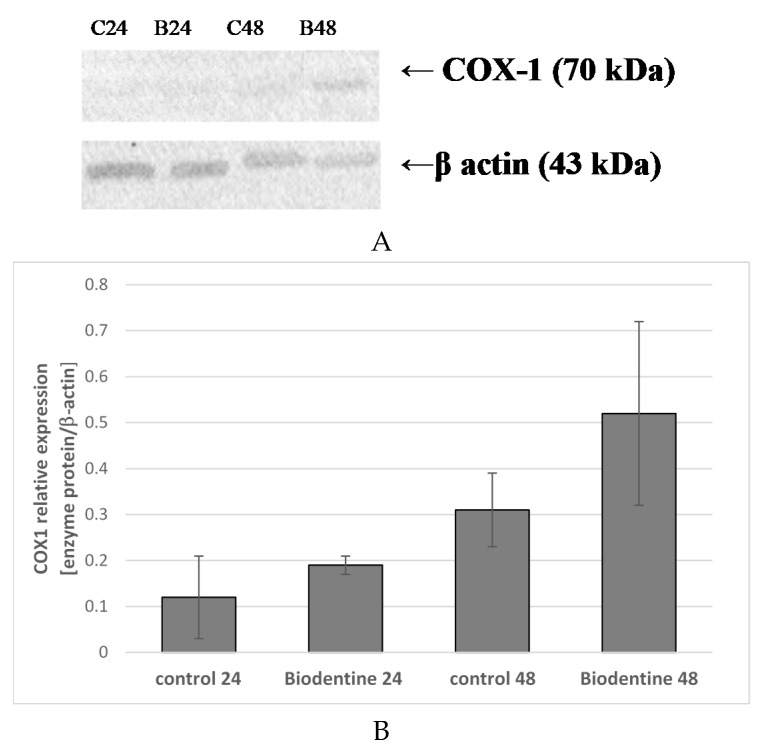
COX-1 protein expression in monocytes. Enzyme protein expression (70 kDa) was determined using the Western blot method. Representative Western blots of COX-1 protein (**A**) and densitometric analysis of COX-1 protein normalized to β-actin (**B**) in monocytes were shown. Cells were cultured for 24 h (B24) and 48 h (B48) in RPMI medium with 10% FBS with Biodentine^TM^, after incubation the cells were harvested by scraping and protein concentration was measured. Control cells were incubated in RPMI medium with 10% Fatal Bovine Serum (FBS) for 24 h as control 24 (C24) or for 48 h as control 48 (C48). Experiments were conducted as three separate assays. There were no statistically significant differences between the studied groups, *p* ≤ 0.5 (U-Mann Whitney test).

**Figure 8 ijms-21-02237-f008:**
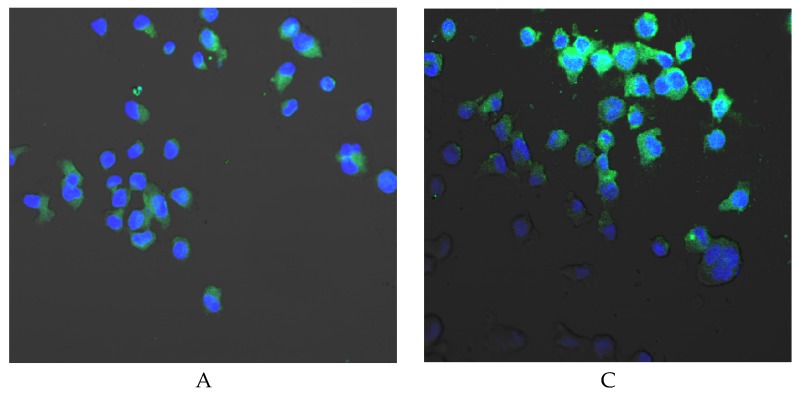
Imaging of COX-1 enzyme expression by fluorescence microscopy in THP-1 monocytes cultured with Biodentine^TM^. Cells were cultured in RPMI medium with 10% FBS—control condition for 24 h (**A**) and for control condition for 48 h (**B**); THP1-monocytes cultured in RPMI medium with 10% FBS with Biodentine^TM^ for 24 h (**C**) and for 48 h (**D**). The immunohistochemistry was performed using specific primary antibody, mouse anti-COX-1 (the overnight incubation at 4 °C) and secondary antibodies conjugated with fluorochrome—anti-mouse IgG-FITC (fluorescein isothiocyanate) (incubation for 45 min at room temperature), enzyme expression visible as green fluorescence indicate COX-1 expression. The nuclei of cells were 4′,6-diamidino-2-phenylindole (DAPI) stained. Image analysis was performed with a confocal microscope using filters 38 HE GFP for green fluorescence and 49 DAPI for blue fluorescence. Objective magnification 20×, computer zoom 2×. Experiments were conducted as six separate assays (each assay in three replicates). No differences in enzymes protein expression were observed, *p* ≤ 0.5 (U Mann-Whitney test).

**Figure 9 ijms-21-02237-f009:**
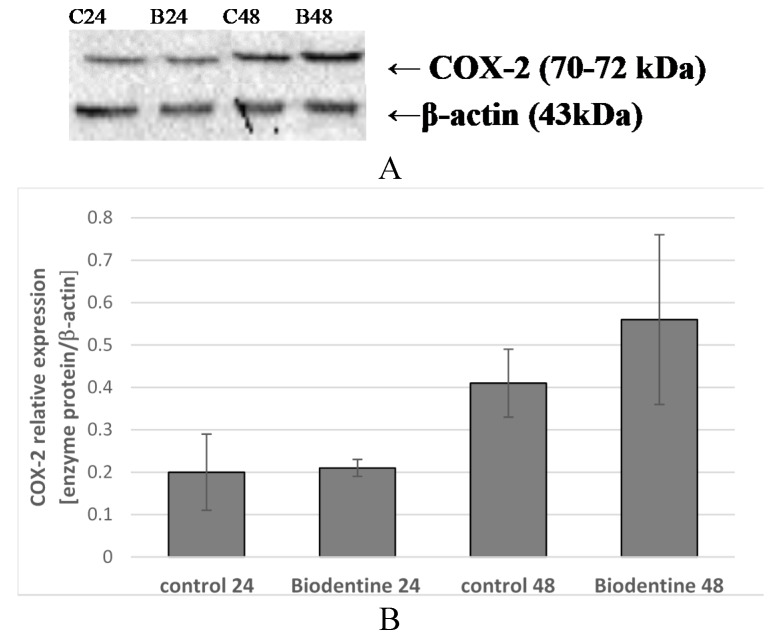
COX-1 protein expression in macrophages. Enzyme protein expression (70 kDa) was determined using Western blot method. Representative Western blots of COX-1 protein (**A**) and densitometric analysis of COX-1 protein normalized to β-actin (**B**) in macrophages were shown. Cells were cultured for 24 h (B24) and 48 h (B48) in RPMI medium with 10% FBS with Biodentine^TM^, after incubation the cells were harvested by scraping and protein concentration was measured. Control cells were incubated in RPMI medium with 10% FBS for 24 h as control 24 (C24) or for 48 h as control 48 (C48). Experiments were conducted as three separate assays. There were no statistically significant differences between the studied groups, *p* ≤ 0.5 (U-Mann Whitney test).

**Figure 10 ijms-21-02237-f010:**
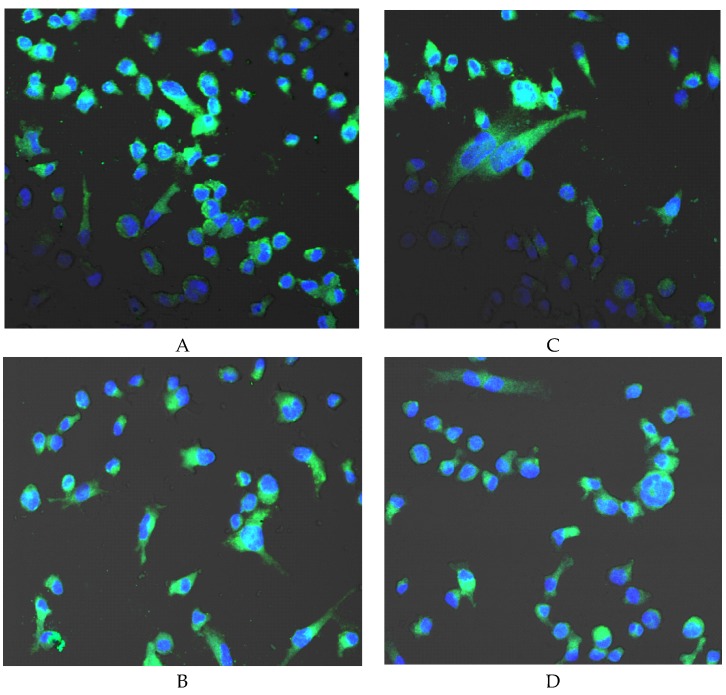
Imaging of COX-1 enzyme expression by fluorescence microscopy in macrophages cultured with Biodentine^TM^. Cells were cultured in RPMI medium with 10% FBS—control condition for 24 h (**A**) and for control condition for 48 h (**B**); macrophages cultured in RPMI medium with 10% FBS with Biodentine^TM^ for 24 h (**C**) and for 48 h (**D**). The immunohistochemistry was performed using specific primary antibody, mouse anti-COX-1 (the overnight incubation at 4 °C) and secondary antibodies conjugated with fluorochrome-anti-mouse IgG-FITC (incubation for 45 min at room temperature), enzyme expression visible as green fluorescence indicate COX-1 expression. The nuclei of cells were DAPI stained. Image analysis was performed with a confocal microscope using filters 38 HE GFP for green fluorescence and 49 DAPI for blue fluorescence. Objective magnification 20×, computer zoom 2×. Experiments were conducted as six separate assays (each assay in three replicates). No differences in enzymes protein expression were observed *p* ≤ 0.5 (U-Mann Whitney test).

**Table 1 ijms-21-02237-t001:** Detailed manufacturer data on the Biodentine^TM^ used for testing.

Preparation	Producer	Composition
Biodentine^TM^	Septodont France	Powder:
		tricalcium and dicalcium silicate
		(3CaO**·**SiO_2_ and 2CaO**·**SiO_2_)
		calcium carbonate (CaCO_3_)
		Zirconium dioxide (ZrO_2_)
		Liquid:
		10% calcium chloride (CaCl_2_**·**2H_2_O)
		water
		polycarboxylate

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
