# Peer review of "Analysis of the Activity and Expression of Cyclooxygenases COX1 and COX2 in THP-1 Monocytes and Macrophages Cultured with BiodentineTM Silicate Cement"

_ijms, 2020, doi:10.3390/ijms21062237_

Round 1

Reviewer 1 Report

The proposed changes have been attended.

Reviewer 2 Report

Dear author,

Congratulations for the paper, well prepared and of great interest to endodontics community.

All the best

This manuscript is a resubmission of an earlier submission. The following is a list of the peer review reports and author responses from that submission.

Round 1

Reviewer 1 Report

The authors present an in vitro study with the aim of showing the influence of Biodentine material over a particular set of inflammatory molecules produced by a monocyte cell line. As detailed below the work has critical problems. My concerns are both technical and conceptual

1- In vitro study, and in consequence, relatively low interest as far as inflammation is a multifactorial process involving an extraordinary complex and diverse molecular pathways. Studying inflammation by the expression of Cox 1 and 2 is, in my view, too reductionist.

2- The right controls should be included in the experiments, no information about the vehicle they are using. A condition in which activation of the Cox expression occurs should be included. A comparison with other bioceramics materials should be added.

3- The source of the macrophages is the very same THP-1 monocyte cell line. No indication of the differentiation is provided, so it is difficult to know whether they actually are working with macrophages further than the indication of the fig2.

4- To our knowledge, the interpretation of the results provided by the authors in the discussion section is not matching with the results presented:

line 299: “In our study, incubation of THP-1 monocytic cells with Biodentine did not cause a significant increase in the concentration of TXB2 or PGE2 in the medium compared to the control”. In my understanding Fig 5 A and B and Fig 6 panel A time 24 h is actually saying the opposite.

Line 301 “The presence of Biodentine did not affect the activation of THP-1 monocytes and their transformation into macrophages”. This is not what appears in fig 2, especially in the caption.

Line 351: “The results of our study showed a lack of macrophage activation and a lack of stimulation of the activity of the tested cyclooxygenase enzymes, and a lack of increase in the concentration of their inflammatory mediators (PGE2 and TXB2) under in vitro incubation with Biodentine. No significant changes in the production of the studied compounds were observed after both 24 and 48 hours of incubation”.

In our interpretation, PGE2 and TXB2 is produced as consequence of the treatment even though Cox1 and 2 do not change. Instead of doge the question, the authors should provide an explanation.

5- We don’t understand why the authors show the statistical significance between the control at different times (fig 5B as an example), in my view is far from the point of the article and, actually suggest a much more complex scenario where the in vitro conditions could be tampering the assay

Minor

1- Many statements poorly or without reference (line 87 is an example)

2- Fig2 They show the differentiation of THP1 cells to macrophages, but they claim activation

3- Fig3 similar number of cells should be seeded, other ways information is difficult to interpret, for instance, comparing panels C and D: the treatment is detrimental for the growth

4- Looking at the caption legends it’s not totally clear to me whether the separate assays are actually independent experiments

5- Fig8 and 9, the authors performed separate experiments, a quantification should be provided to claim no differences

6- Fig 9 at first view in immunostaining no differences are along time, which is not the case when using western blot

Reviewer 2 Report

Dear authors,

About your paper I am sending some comments:

Line 17: I advice the word biochemical.

Line 19: Change: The aim of our study, to: the aim of this study…

Line 38: I advice the word bioceramic.

Line 112: I advice: the aim of this study…Biodentine a silicate based material…

Fig. 2, 3, 5, 6, 7, 8, 9, 10: Improve the legend. Remove the text refered in the materials and methods.

Line 290: Could you please elaborate a few lines on advances of biomaterials used for treating pulp wounds. The below paper help you with this:

Anabela Baptista Paula, Mafalda Laranjo, CMiguel Marto, Ana Abrantes, João Casalta-Lopes, ACristina Gonçalves, Ana Sarmento, Manuel Ferreira, MFilomena Botelho, Eunice Carrilho. Biodentine™ boosts, WhiteProRoot®MTA increases and Life® suppresses odontoblast activity in pulp capping therapies. Materials 2019, 12, 1184; doi:10.3390/ma12071184

Reviewer 3 Report

The study conducted by these authors is relevant and provides answers on the anti-inflammatory role of a material based on tricalcium silicate. the use of Biodentine as bioactive material in many applications is very common. so these new data will serve to the better understanding how this material works in the regeneration of the pulp dentine complex context.

The authors search to understand how biodentine could may affect the expression of the enzymes COX1 and COX2 in THP-1 monocytes/macrophages, and the level of PGE2 and TXB2 

We must salute the original work of the team since they worked for the first time on THP-1 monocyte/macrophage cell model with a  biomaterial in direct contact with cells.

Several bibliographic references are missing in the introduction section (p 3, line 77-90.

Figure 1 is not entirely correct. Concerning apexification, it should be interesting to draw an immature apex to be realistic. In the same way, the external and internal resrption must not be at the same location.

in the material and methods section, why the protein concentration appears after Western blot.

In the results section: it is not easy to follow all results. furthermore thera are few results with statistical significant. It is not so easy to conclude. Is it possible to increase number of samples.

in the discussion section there is a lot of non relevant informations p 12, line 317 to 350. Please rewrite this section.

in general please write in vitro in vitro and biodentine, Biodentine TM

Please be less assertive in the conclusions section

Responses to the Review Report

The authors present an in vitro study with the aim of showing the influence of Biodentine material over a particular set of inflammatory molecules produced by a monocyte cell line. As detailed below the work has critical problems. My concerns are both technical and conceptual

1- In vitro study, and in consequence, relatively low interest as far as inflammation is a multifactorial process involving an extraordinary complex and diverse molecular pathways. Studying inflammation by the expression of Cox 1 and 2 is, in my view, too reductionist.

We welcome the Reviewer’s suggestions, however we find it difficult to adjust the paper to the recommendation, as well as have some problems understanding its validity. The in vitro model used in the paper has previously been used by numerous authors to directly analyse the effect of various factors on the expression and activity of cyclooxygenases and their role in development and increase of inflammation. Moreover, the THP-1 cellular line monocytes and macrophages obtained from them constitute a widely applied cellular model used to investigate immune reaction. It was proven that the results of studies conducted with these cells can be applicable with respect to the human organism. THP-1 show similarities to the human monocytes/macrophages both in terms of morphology as well as function. The literature on the subject considers them to be an appropriate model for the study of monocyte/macrophage response, for the purpose of determining “macrophage differentiation’, as well as investigating the effect of external factors on macrophages [Chanput W, Mes JJ, Wichers HJ. THP-1 cell line: an in vitro model for immunomodulation approach. Int Immunopharmacol 2014, 23, 37-45].

This model was employed, as well as referred to, in numerous publications, just to list a few:

Lund ME et al. The choice of phorbol 12-myristate 13-acetate differentiation protocol influences the response of THP-1 macrophages to a pro-inflammatory stimulus. J Immunol Methods. (2016).

 Chanput W et al. Transcription profiles of LPS-stimulated THP-1 monocytes and macrophages: a tool to study inflammation modulating effects of food-derived compounds. Food Funct. (2010)  

Olszowski T, Gutowska I, Baranowska-Bosiacka I, Łukomska A, Drozd A, Chlubek D. Cadmium Alters the Concentration of Fatty Acids in THP-1 Macrophages. Biol Trace Elem Res. 2018 Mar;182(1):29-36. doi: 10.1007/s12011-017-1071-6.

 Olszowski T, Baranowska-Bosiacka I, Gutowska I, Piotrowska K, Mierzejewska K, Korbecki J, Kurzawski M, Tarnowski M, Chlubek D. The Effects of Cadmium at Low Environmental Concentrations on THP-1 Macrophage Apoptosis. Int J Mol Sci. 2015 Sep 7;16(9):21410-27. doi: 10.3390/ijms160921410.

 Olszowski T, Gutowska I, Baranowska-Bosiacka I, Piotrowska K, Korbecki J, Kurzawski M, Chlubek D. The Effect of Cadmium on COX-1 and COX-2 Gene, Protein Expression, and Enzymatic Activity in THP-1 Macrophages. Biol Trace Elem Res. 2015 Jun;165(2):135-44. doi: 10.1007/s12011-015-0234-6.

 Sikora M, Baranowska-Bosiacka I, Łukomska A, Goschorska M, Chlubek D. Expression of metalloproteinase 2 (MMP-2) and metalloproteinase 9 (MMP-9) in THP-1 macrophages cultured with three-dimensional titanium mini-plate systems used for surgical treatment of condylar fractures.Acta Biochim Pol. 2019 Jul 31;66(3):291-298. doi: 10.18388/abp.2019_2766.

 Korbecki J, Gutowska I, Wiercioch M, Łukomska A, Tarnowski M, Drozd A, Barczak K, Chlubek D, Baranowska-Bosiacka I. Sodium Orthovanadate Changes Fatty Acid Composition and Increased Expression of Stearoyl-Coenzyme A Desaturase in THP-1 Macrophages.Biol Trace Elem Res. 2020 Jan;193(1):152-161. doi: 10.1007/s12011-019-01699-2.

 Baranowska-Bosiacka I, Olszowski T, Gutowska I, Korbecki J, Rębacz-Maron E, Barczak K, Lubkowska A, Chlubek D. Fatty acid levels alterations in THP-1 macrophages cultured with lead (Pb).J Trace Elem Med Biol. 2019 Mar;52:222-231. doi: 10.1016/j.jtemb.2019.01.003.

 Goschorska M, Gutowska I, Baranowska-Bosiacka I, Piotrowska K, Metryka E, Safranow K, Chlubek D. Influence of Acetylcholinesterase Inhibitors Used in Alzheimer's Disease Treatment on the Activity of Antioxidant Enzymes and the Concentration of Glutathione in THP-1 Macrophages under Fluoride-Induced Oxidative Stress. Int J Environ Res Public Health. 2018 Dec 20;16(1).

Goschorska M, Baranowska-Bosiacka I, Gutowska I, Tarnowski M, Piotrowska K, Metryka E, Safranow K, Chlubek D. Effect of acetylcholinesterase inhibitors donepezil and rivastigmine on the activity and expression of cyclooxygenases in a model of the inflammatory action of fluoride on macrophages obtained from THP-1 monocytes. Toxicology. 2018 Aug 1;406-407:9-20. doi: 10.1016/j.tox.2018.05.007.

 Sikora M, Goschorska M, Baranowska-Bosiacka I, Chlubek D. In Vitro Effect of 3D Plates Used for Surgical Treatment of Condylar Fractures on Prostaglandin E₂ (PGE₂) and Thromboxane B₂ (TXB₂) Concentration in THP-1 Macrophages. Int J Mol Sci. 2017 Dec 8;18(12). pii: E2638. doi: 10.3390/ijms18122638.

2- The right controls should be included in the experiments, no information about the vehicle they are using.

The said information was stated in Materials and Methods, section 4.4 Verification of BiodentineTM induced activation of THP-1 monocytes and initiation of inflammatory reaction

“In the first experiment, THP-1 cells were cultured for 24 and 48 hours in RPMI medium with 10% FBS in the presence of BiodentineTM prepared according to the manufacturer’s instructions”.

The subsequent experiments were conducted under the same conditions, i.e. Biodentine was not dissolved in any vehicle, only in RPMI medium. The cells under controlled conditions were cultured in a standard medium, the detailed composition is given in Materials and Methods: 4.3 Cell culture and treatment: „The experiments were conducted on human macrophages obtained from THP-1 cells. The cells were cultured in Roswell Park Memorial Institute (RPMI) 1640 (Sigma, St. Louis, MO), supplemented with 100 IU/ml penicillin and 10 µg/ml streptomycin (Life Technologies, Inc., GrandIsland, NY) in the presence of 10% thermally inactivated fetal bovine serum (FBS, LifeTechnologies). Cells were cultured in a humid atmosphere at 37°C in 5% CO2, the medium was refreshed every 48 hours. Prior to the experiment, THP-1 cells were placed in culture flasks at an initial density of 2.5 x 105 cells/well (Corning, Cambridge, MA). The differentiation of THP-1 cells in the macrophages was achieved by administration of 100 nM PMA for 24 hours [49]”.

 A condition in which activation of the Cox expression occurs should be included.

We verified whether Biodentine causes direct activation of monocytes in the experiment 4.4 Verification of BiodentineTM induced activation of THP-1 monocytes and initiation of inflammatory reaction a następnie w kolejnym : 4.5 The differentiation of THP-1 cells into macrophages. Flow cytometry measurement” and, in the section Materials and Methods, the experimental conditions are described in detail.

A comparison with other bioceramics materials should be added.

According to the Reviewer’s suggestion, we added the fragment concerning other bioceramic materials.

3- The source of the macrophages is the very same THP-1 monocyte cell line. No indication of the differentiation is provided, so it is difficult to know whether they actually are working with macrophages further than the indication of the fig2.

In Materials and Methods, section 4.3 Cell culture and treatment, we provide a description of transforming THP-1 monocytes into macrophages i.e., . „The differentiation of THP-1 cells in the macrophages was achieved by administration of 100 nM PMA for 24 hours” Auwerx, J. The human leukemia cell line, THP1: A multifacetted model for study of monocyte-macrophage differentiation. Experientia 1991, 47, 22–31.

In Fig.2 we provide additional explanation and show flow cytometry: CD68 expression (as a marker for macrophage differentiation) increased after 48 h treatment with PMA (200nM) and expression of CD14 (marker for monocyte differentiation) did not change.

4- To our knowledge, the interpretation of the results provided by the authors in the discussion section is not matching with the results presented:

line 299: “In our study, incubation of THP-1 monocytic cells with Biodentine did not cause a significant increase in the concentration of TXB2 or PGE2 in the medium compared to the control”. In my understanding Fig 5 A and B and Fig 6 panel A time 24 h is actually saying the opposite.

We are very grateful for this remark. According to the Reviewer’s remark we corrected the sentence:

“In our study, incubation of THP-1 monocytic cells with BiodentineTM did not cause a significant increase in the concentration of TXB2 or PGE2 in the medium compared to the cells incubated with control (after 48 hours of incubation)”.

Line 301 “The presence of Biodentine did not affect the activation of THP-1 monocytes and their transformation into macrophages”. This is not what appears in fig 2, especially in the caption.

The results shown in Fig.2 (flow cytometry analysis) show that “THP-1 monocytic cells were incubated in the presence of a Biodentine without PMA (upper quadrants) and treated with PMA (lower quadrants). Expression was determined by flow cytometry. CD68 expression (marker for macrophage differentiation) increased after 48 h treatment with PMA (200nM) as compared to non-treated cells. Expression of CD14 (marker for monocyte differentiation) did not change significantly (p=0.65).

According to the Reviewer’s suggestion, to clarify any doubts, we added the result of the statistical analysis: did not change significantly (p=0.65).  

Line 351: “The results of our study showed a lack of macrophage activation and a lack of stimulation of the activity of the tested cyclooxygenase enzymes, and a lack of increase in the concentration of their inflammatory mediators (PGE2 and TXB2) under in vitro incubation with Biodentine. No significant changes in the production of the studied compounds were observed after both 24 and 48 hours of incubation”.

In our interpretation, PGE2 and TXB2 is produced as consequence of the treatment even though Cox1 and 2 do not change. Instead of doge the question, the authors should provide an explanation.

The sentence was changed in line with the Reviewer’s suggestion: “The results of our study showed that macrophages incubated with Biodentine in vitro in 48 hours did not show activation characteristics. Moreover, no significant increase in the expression of cyclooxygenase or significant increase in inflammation mediators (PGE2 and TXB2) were found. This indicates that Biodentine, under the conditions presented in the paper, does not initiate inflammation and does not contribute its propagation.”

5- We don’t understand why the authors show the statistical significance between the control at different times (fig 5B as an example), in my view is far from the point of the article and, actually suggest a much more complex scenario where the in vitro conditions could be tampering the assay

We demonstrate a significant increase between control 24 and control 48 since, as was stated in the Introduction,: “COX-1 has been considered a constitutive enzyme which does not play a significant role in the inflammatory process [28]. However, it is now known that in some tissues this enzyme is inducible and plays a role in the initial phase of the response to the factors initiating prostanoid synthesis. COX-1 is a source of PGE2 and TXA2 (which is metabolized to the more stable TXB2). COX-2 is an enzyme subject to inducible expression in response to e.g. pro-inflammatory cytokines or cytokines produced by cell growth factors (including monocytes) and plays a dominant role in chronic inflammation [29, 30].

This means that even under the control conditions the cells can synthesise these enzymes, therefore we present both COX and COX2 expression levels as well as level of the activity products after 24 ans 48 hours.

Minor

1- Many statements poorly or without reference (line 87 is an example)

The references were added according to the Reviewer’s suggestions. 

2- Fig2 They show the differentiation of THP1 cells to macrophages, but they claim activation

We agree that Fig.2 shows differentiation of THP-1 monocytes to macrophages, which however states not only their differentiation but, more importantly, activation of macrophages – evidenced by expression of specific receptors. Therefore, we decide not to change the description of Fig.2.

3- Fig3 similar number of cells should be seeded, other ways information is difficult to interpret, for instance, comparing panels C and D: the treatment is detrimental for the growth

Obviously we also agree with the Reviewer’s suggestion that the number of cells affects the result of the experiment, and that is why the cells were always sown in the same number – as is stated in Material and Methods section: “THP-1 cells were placed in culture flasks at an initial density of 2.5 x 105 cells/well”.

4- Looking at the caption legends it’s not totally clear to me whether the separate assays are actually independent experiments

Experiments were conducted as six separate assays (each assay in three replicates), which means n=18.

5- Fig8 and 9, the authors performed separate experiments, a quantification should be provided to claim no differences

According to the Reviewer’s suggestion we added p value. There weren’t any statistically significant differences between the studied group, p≤0.5 (U-Mann Whitney test).

6- Fig 9 at first view in immunostaining no differences are along time, which is not the case when using western blot

The presented picture is only a representation, the analysis was quantitatively confirmed with a computer software using a microscope allowing to determine quantitatively the intensity of fluorescence which, in turn, allows to conduct the statistical analysis which showed no statistical differences.

Review 2

Dear authors, About your paper I am sending some comments:

Line 17: I advice the word biochemical.

We corrected the sentence according to the Reviewer’s remark

Line 19: Change: The aim of our study, to: the aim of this study…

We corrected the sentence according to the Reviewer’s remark

Line 38: I advice the word bioceramic.

We corrected the sentence according to the Reviewer’s remark

Line 112: I advice: the aim of this study…Biodentine a silicate based material…

We corrected the sentence according to the Reviewer’s remark

Fig. 2, 3, 5, 6, 7, 8, 9, 10: Improve the legend. Remove the text refered in the materials and methods.

We thank the Reviewer for the suggestion, however we would like to keep the present form of the legend as it allows a quick overview of the results of the paper and even without the detailed analysis of the whole Material and Methods section. We would kindly ask to accept such a form.

Line 290: Could you please elaborate a few lines on advances of biomaterials used for treating pulp wounds. The below paper help you with this:

Anabela Baptista Paula, Mafalda Laranjo, CMiguel Marto, Ana Abrantes, João Casalta-Lopes, ACristina Gonçalves, Ana Sarmento, Manuel Ferreira, MFilomena Botelho, Eunice Carrilho. Biodentine™ boosts, WhiteProRoot®MTA increases and Life® suppresses odontoblast activity in pulp capping therapies. Materials 2019, 12, 1184; doi:10.3390/ma12071184

According to the Reviewer’s suggestion, the fragment was supplemented and corrected.

Review 3

Comments and Suggestions for Authors

The study conducted by these authors is relevant and provides answers on the anti-inflammatory role of a material based on tricalcium silicate. the use of Biodentine as bioactive material in many applications is very common. so these new data will serve to the better understanding how this material works in the regeneration of the pulp dentine complex context.

The authors search to understand how biodentine could may affect the expression of the enzymes COX1 and COX2 in THP-1 monocytes/macrophages, and the level of PGE2 and TXB2 

We must salute the original work of the team since they worked for the first time on THP-1 monocyte/macrophage cell model with a  biomaterial in direct contact with cells.

Several bibliographic references are missing in the introduction section (p 3, line 77-90.

References were added according to the Reviewer’s suggestions.

Figure 1 is not entirely correct. Concerning apexification, it should be interesting to draw an immature apex to be realistic. In the same way, the external and internal resrption must not be at the same location.

Figure 1 has been corrected according to the Reviewer’s remark.

In the material and methods section, why the protein concentration appears after Western blot.

We express our thanks for the Reviewer’s suggestions. The section Protein concentration appears at the end of the said section since it refers not only to WB analysis but also to analyses made with ELISA – all the results were calculated to 1 mg/protein. Therefore, we would like to keep this section in the present form, so as to avoid repeating the method of protein determination in all sections describing the analyses.

In the results section: it is not easy to follow all results. furthermore thera are few results with statistical significant. It is not so easy to conclude. Is it possible to increase number of samples.

The Results section was shortened. The number of samples (n=6) was sufficient to conduct a correct statistical analysis (U Mann-Whitney test) and, in our view, there is no need to increase their number as it does not have any effect on test strength.

In the discussion section there is a lot of non relevant informations p 12, line 317 to 350. Please rewrite this section.

Discussion was shortened in the indicated fragment.

In general please write in vitro in vitro and biodentine, Biodentine TM

Please be less assertive in the conclusions section